# *Strongyloides stercoralis* infection induces gut dysbiosis in chronic kidney disease patients

**Nguyen Thi Hai**[1,2,3], **Nuttanan Hongsrichan**[1,3], **Kitti Intuyod**[3,4], **Porntip Pinlaor**[3,5], **Manachai Yingklang**[3,6], **Apisit Chaidee**[1,3], **Thatsanapong Pongking**[3,7], **Sirirat Anutrakulchai**[3,8], **Ubon Cha'on**[3,9], **Somchai Pinlaor** [1,3]*

**1** Department of Parasitology, Faculty of Medicine, Khon Kaen University, Khon Kaen, Thailand, **2** Department of Parasitology, Faculty of Basic Medicine, Thai Nguyen University of Medicine and Pharmacy, Thai Nguyen, Vietnam, **3** Chronic Kidney Disease Prevention in the Northeastern Thailand, Khon Kaen, Thailand, **4** Department of Pathology, Faculty of Medicine, Khon Kaen University, Khon Kaen, Thailand, **5** Centre for Research and Development of Medical Diagnostic Laboratories, Faculty of Associated Medical Sciences, Khon Kaen University, Khon Kaen, Thailand, **6** Department of Fundamentals of Public Health, Faculty of Public Health, Burapha University, Chonburi 20131, Thailand, **7** Science Program in Biomedical Science, Khon Kaen University, Khon Kaen, Thailand, **8** Department of Medicine, Faculty of Medicine, Khon Kaen University, Khon Kaen, Thailand, **9** Department of Biochemistry, Faculty of Medicine, Khon Kaen University, Khon Kaen, Thailand

\* psomec@kku.ac.th

**Data Availability Statement:** The Illumina datasets for the profile of gut microbiota (Raw data) are available from Mendeley Data (DOI: 10.17632/hvbvrtc34x.1).

## Abstract

### Background

*Strongyloides stercoralis* infection typically causes severe symptoms in immunocompromised patients. This infection can also alter the gut microbiota and is often found in areas where chronic kidney disease (CKD) is common. However, the relationship between *S. stercoralis* and the gut microbiome in chronic kidney disease (CKD) is not understood fully. Recent studies have shown that gut dysbiosis plays an important role in the progression of CKD. Hence, this study aims to investigate the association of *S. stercoralis* infection and gut microbiome in CKD patients.

### Methodology/Principal findings

Among 838 volunteers from Khon Kaen Province, northeastern Thailand, 40 subjects with CKD were enrolled and divided into two groups (*S. stercoralis*-infected and -uninfected) matched for age, sex and biochemical parameters. Next-generation technology was used to amplify and sequence the V3-V4 region of the 16S rRNA gene to provide a profile of the gut microbiota. Results revealed that members of the *S. stercoralis*-infected group had lower gut microbial diversity than was seen in the uninfected group. Interestingly, there was significantly greater representation of some pathogenic bacteria in the *S. stercoralis*-infected CKD group, including *Escherichia-Shigella* ($P$ = 0.013), *Rothia* ($P$ = 0.013) and *Aggregatibacter* ($P$ = 0.03). There was also a trend towards increased *Actinomyces*, *Streptococcus* and *Haemophilus* ($P$ > 0.05) in this group. On the other hand, the *S. stercoralis*-infected CKD group had significantly lower representation of SCFA-producing bacteria such as *Anaerostipes* ($P$ = 0.01), *Coprococcus*_1 (0.043) and a non-significant decrease of *Akkermansia*, *Eubacterium rectale* and *Eubacterium hallii* ($P$ > 0.05) relative to the uninfected group. Interesting,

**Funding:** This study was supported by Research and Graduate Studies, Khon Kaen University (RP65-2-001 to S.P.). N.T.H. received the Invitation research grant, Faculty of Medicine Research Grant (IV63136), Khon Kaen University. S.A. acknowledged research funds from Thailand Science Research and Innovation (TSRI), through Program Management Unit for Competitiveness (PMUC), number C10F630030 and CKDNET (grant no. CKDNET2559007).The funders had no role in study design, data collection and analysis, decision to publish, or preparation of the manuscript.

**Competing interests:** The authors have declared that no competing interest exist.

the genera *Escherichia-Shigella* and *Anaerostipes* exhibited opposing trends, which were significantly related to sex, age, infection status and CKD stages. The genus *Escherichia-Shigella* was significantly more abundant in CKD patients over the age of 65 years and infected with *S. stercoralis*. A correlation analysis showed inverse moderate correlation between the abundance of the genus of *Escherichia-Shigella* and the level of estimated glomerular filtration rate (eGFR).

## Conclusions/Significance

Conclusion, the results suggest that *S. stercoralis* infection induced gut dysbiosis in the CKD patients, which might be involved in CKD progression.

### Author summary

Human strongyloidiasis is caused by a soil-transmitted helminth, *Strongyloides stercoralis*, which typically causes severe symptoms in immunocompromised individuals. This infection can also alter the gut microbiota and is often found in areas where chronic kidney disease (CKD) is common. However, the relationship between *S. stercoralis* and the gut microbiome in CKD is not known. This is the first study to investigate the gut microbiota of CKD patients with and without *S. stercoralis* using high-throughput sequencing of the V3–V4 region of the 16S rRNA gene. Infection with *S. stercoralis* was associated with reduced gut microbial diversity. In addition, infection with this nematode led to reduced abundance of SCFA-producing bacteria and enrichment of pathogenic bacteria. In particular, there were significant differences in abundance of the beneficial genus *Anaerostipes* (a decrease) and the pathogenic taxon *Escherichia-Shigella* (an increase) in CKD patients infected with *S. stercoralis* relative to controls. In the infected group, the representation of *Escherichia-Shigella* was significantly higher in patients over the age of 65 years. There was a significant inverse moderate correlation of *Escherichia-Shigella* with the estimated glomerular filtration rate (eGFR).

## Introduction

An imbalance within the microbiota in the gastrointestinal tract, termed gut dysbiosis, contributes to the development and progression of many diseases including chronic kidney disease (CKD) [1]. Many studies have shown a significant difference in the abundance of bacterial populations in the gastrointestinal tract (GI) between CKD and control individuals. Substantially lower proportions of *Bifidobacterium*, *Lactobacillaceae*, *Bacteroidaceae* and *Prevotellaceae* were seen in CKD patients, including those undergoing hemodialysis, while the proportions of *Enterobacteriaceae*, especially *Enterobacter*, *Klebsiella* and *Escherichia*, were notably higher [2–5]. The production of uremic toxins (indoxyl sulphate (IS), trimethylamine-N-oxide (TMAO)), which results from nutrient processing by gut microbiota, and the reduction of fiber-derived short-chain fatty acids, are linked with CKD progression [6–8]. Recent studies have found various factors involved in microbial dysbiosis and CKD, such as the use of antibiotics [9], decreased consumption of dietary fiber [10], and oral iron intake [11]. However, many etiological factors associated with CKD remain obscure [12], particularly those due to infection with intestinal parasites.

The ability of GI parasitic infection to change the gut microbiota and host-microbiota interactions has been clearly identified. Infection with intestinal parasites either induced gut dysbiosis or provided protection against dysbiosis and inflammatory disease [13]. *Strongyloides stercoralis* is one of the most medically important parasites in northeastern Thailand, where the prevalence of CKD is also high [14]. Typically, *S. stercoralis* infection causes only mild GI symptoms. However, when immunity is suppressed by, for example, CKD or HIV infection, the parasite can rapidly multiply leading to hyperinfection and disseminated strongyloidiasis, which is a life-threatening condition [15–18].

Recent studies have demonstrated that *S. stercoralis* induces an increase in bacterial diversity and changes faecal microbiota [19,20]. By using metagenomic analysis, microbial alpha diversity was found to increase and beta diversity decrease, in the faecal microbial profiles of *S. stercoralis*-infected individuals compared to uninfected. Faecal metabolite analysis detected marked increases in the abundance of selected amino acids and decrease in short-chain fatty acids in *S. stercoralis* infection, relative to uninfected controls [20]. Taken together, we therefore hypothesize that *S. stercoralis* infection changes the gut microbiome, contributing to progression of CKD.

To test this hypothesis, metagenomic analysis was done in patients with CKD to investigate the changes in the gut microbiota that can be attributed to *S. stercoralis* infection. The result from this study might be useful for identifying strategies to limit development and progression of chronic kidney disease.

## Methods

### Ethics statement

The human ethical review committee of Khon Kaen University (HE631200) approved the protocol of study. Informed consent was obtained from all participants under the CKD project and was verbal or written [14].

### Study population

The study was conducted between January 2017 and May 2018 at Donchang sub-district, Khon Kaen Province, northeastern Thailand as a part of the Chronic Kidney Disease Northeastern Thailand (CKDNET) project. Included in the study were individuals (>35 years of age) with chronic kidney disease. Their diagnosis, done by a nephrologist [14], included clinically proven impaired kidney structure or renal function, as detected using ultrasonography, and a finding of reduced eGFR. The staging (stages 1 to 5) based on the estimated glomerular filtration rate (eGFR) [21,22] was estimated for each individual. In patients with eGFR > 60 ml/min/1.73 m2, kidney damage was confirmed based on urine albumin-to-creatinine ratio (UACR), hematuria and abnormal renal ultrasound. Stool examination was performed on CKD patients using the modified formalin ethyl acetate concentration technique (FECT) and modified agar plate culture (mAPC) as previously reported [23].

Exclusion criteria were as follows: use of antibiotics or probiotics, diabetes, autoimmune disease, urinary tract infection and infection with intestinal parasites other than *S. stercoralis*. Twenty CKD patients with *S. stercoralis* infection (Ss+) met these criteria and were included. The control group consisted of 20 CKD patients free of *S. stercoralis* infection (Ss-) who otherwise matched the characteristics (including sex, age and biochemical factors) of the Ss+ group (Table 1). These datasets were obtained from the medical records of CKDNET and from a recent study [23]. Absence of *S. stercoralis* infection in members of the control group was also confirmed using PCR tests.

**Table 1. Characteristics of chronic kidney disease patients.**

| Parameters | Normal range | Ss- (n = 20) | Ss+ (n = 20) | *P* value |
|---|---|---|---|---|
| Sex | Male | 12 | 12 | |
| | Female | 8 | 8 | |
| Age years | | 64.60±11.3 | 64.85±13.4 | 0.95[a] |
| BMI kg/m$^2$ | 18-25 | 23.6±3.8 | 22.14±4.1 | 0.26[a] |
| MCV fL | 79.0-94.8 | 80.53±8.9 | 83.45±9.7 | 0.33[a] |
| MCH pg | 25.6-32.2 | 26.25±3.7 | 27.48±3.7 | 0.30[a] |
| MCHC g/dL | 32.2-36.5 | 32.49±1.2 | 32.84±0.9 | 0.29[a] |
| eGFR ml/min/1.73 m$^2$ | >=90 | 73.3±20.3 | 71.3±25.4 | 0.78[a] |
| Neutrophil (NE)% | 50-70 | 52.00±7.0 | 48.50±13.6 | 0.3[a] |
| Lymphocyte (LY)% | 20-40 | 31.99±6.6 | 33.42±10.0 | 0.60[a] |
| Monocyte (MO)% | 2-8 | 7.56±1.8 | 7.27±1.9 | 0.63[a] |
| Eosinophil (EO)% | 1-3 | 7.60±4.5 | 9.98±8.7 | 0.62[b] |
| Basophils (BA) % | 0-1 | 0.78±0.4 | 0.84±0.5 | 0.84[b] |
| Glucose mg/dL | 70-100 | 89.3±11.2 | 90.45±13.0 | 0.90[b] |
| LDL Cholesterol mg/dL | 10-150 | 121.0±34.4 | 114.75±28.4 | 0.94[b] |
| Hemoglobin g/dL | 13.0-16.7 | 12.18±2.4 | 12.44±1.8 | 0.70[a] |
| Hematocrit % | 34-51 | 37.8±5.7 | 37.7±4.9 | 0.95[a] |
| Uric acid mg/dL | 2.7-7.0 | 6.14±1.6 | 5.85±1.6 | 0.57[a] |
| Urine creatinine mg/dL | 25-400 | 128.89±63.2 | 114.82±61.7 | 0.43[b] |
| Microalbumin mg/dL | 0.2-1.9 | 6.90±18.3 | 5.55±6.2 | 0.29[b] |
| UACR mg/g | <30 | 50.02±114.8 | 140.43±229.2 | 0.096[b] |
| Hemoglobin A1c % | 4.6-6.5 | 5.50±0.6 | 5.45±0.5 | 0.75[a] |
| Bpsys mmhg | <140 | 125.94±17.7 | 129.56±16.5 | 0.54[a] |
| Bpdia mmhg | <90 | 81.38±7.87 | 76.33±9.43 | 0.46[a] |

Data are presented as mean ± standard deviation of the mean. Independent t-tests (a) and Mann-Whitney U tests (b) were used to calculate *P* values.

Abbreviations: BMI, body mass index; Bpsys mmhg, Blood pressure systolic (mmhg); Bpdia mmhg, Blood pressure diastolic (mmhg); MCV, Mean corpuscular volume; MCH, Mean Corpuscular Hemoglobin; MCHC, mean corpuscular hemoglobin concentration; U, urine; UACR, urine albumin-to-creatinine ratio.

## Sample collection, DNA extraction

Faecal and serum samples (n = 40) were collected from the CKD Northeast Thailand project and kept at -20°C until analyzed. DNA was extracted from faecal samples using the QIAamp Kit (Qiagen, Germany). A Nanodrop 2000 spectrophotometer (NanoDrop Technologies, Wilmington, DE, USA) was used to measure DNA concentration and 1.5% agarose gel electrophoresis was used to check the DNA quality.

## 16S rRNA gene sequencing and analysis

To confirm that V3-V4 regions (about 450-500 bp) of the prokaryotic 16S rRNA gene could be amplified, PCR was used to amplify DNA from each faecal sample. Specific V3-V4 primers were used, V3-forward: 5'-CCTACGGGNGGCWGCAG and V4-reverse: 5'-TACNVGGG-TATCTAATCC [24]. Each reaction (20.1 μL) contained 2 μL of 10x buffer MgCl$_2$, 0.4 μL of 50mM MgCl$_2$, 0.6 μL of 10mM dNTP, 1 μL of 5 μM of each primer, 0.1μL Platinum *Taq* DNA polymerase and distilled water. The amplification profile was initial denaturation at 94°C for 5 min, at 94°C for 40 sec, then 35 cycles of 52.8°C for 30 sec, 72°C for 2 min, followed by a final extension at 72°C for 10 min. PCR product was electrophoresed in a 1.5% agarose gel to confirm the presence of a band of the expected size.

A sequencing library was generated for each sample using NEBNext Ultra DNA Library Prep Kit for Illumina (Thermo Scientific) following the manufacturer's recommendations. The library quality was assessed on the Qubit@ 2.0 Fluorometer (Thermo Scientific) and Agilent Bioanalyzer 2100 system. Finally, the library was sequenced on an Illumina platform and 250 bp paired-end reads were generated. Processing and quality control of these reads used the following steps: 1) Data split (based on their unique barcode) and truncation by cutting off the barcode and primer sequences; 2) Sequence assembly (paired-end reads were merged using FLASH [25] to generate raw tags; 3) Data filtration (quality filtering on the raw tags was performed to obtain high-quality clean tags [26] according to the QIIME(V1.7.0, http://qiime.org/index.html) quality-control process [27]; 4) Chimera removal (the tags were compared with a reference database [28] using the UCHIME algorithm [29] to detect chimera sequences, which were then removed [30].

Sequences sharing ≥97% similarity were assigned to the same operational taxonomic unit (OTU) by using Uparse v7.0.1001 [31] and species annotated by reference to the GreenGene Database [32] based on RDP 3 classifier [33] algorithm. The sequence alignment was conducted using the MUSCLE software (Version 3.8.31) [34]. Information on abundance of each OTU was normalized relative to the sample with the fewest sequences. Subsequent analyses of alpha diversity were performed based on this normalized data. Alpha diversity indicates species diversity for a sample using six indices (observed-species, Chao1, Shannon, Simpson, ACE and Good's coverage). All these indices were calculated using QIIME (Version 1.7.0) and displayed with the help of R software (Version 2.15.3). To show beta diversity, weighted and unweighted UniFrac metrics were used to evaluate differences of samples in species complexity by using QIIME software (Version 1.7.0). Principal Coordinate Analysis (PCoA) was performed to visualize complex and multidimensional data. PCoA analysis was done using the WGCNA package, stat packages and ggplot2 package in R software (Version 2.15.3). Metastats was used to detect taxa with significant intra-group variation. The potential biomarkers were detected by using LEfSe (linear discriminant analysis (LDA) Effect Size) [35]. Raw data are available from Mendeley Data (DOI: 10.17632/hvbvrtc34x.1).

## Polymerase Chain Reaction (PCR)

Conventional PCR was used to confirm the absence of *S. stercoralis* infection in members of the control group (n = 20). Primers were designed specifically to amplify a 125-bp fragment from a *S. stercoralis* dispersed repetitive sequence, GenBank: AY08262 [36]. The forward primer was SSC-F 5′ CTCAGCTCCAGTAAAGCAACAG 3′ and reverse primer was SSC-R 5′ AGCTGAATCTGGAGAGTG AAGA 3′. PCR amplification was performed in a 12.5 μL volume with Dream *Taq* PCR Master-mix (Thermo Fisher Scientific, Vilnius, Lithuania), 1 μL of each primer, 5 μL of 9–155 ng/μL DNA, and PCR-grade water. The amplification profile was initial denaturation at 95°C for 10 min, followed by 35 cycles of 95°C for 1 min, 60°C for 1 min 30 s and 72°C for 1 min; then a final extension at 72°C for 10 min. To confirm amplification and amplicon size, the PCR products were resolved on a 2% agarose gel stained with ethidium bromide.

## Statistical analysis

Statistical analyses, including ANOVA, independent t-test, Welch's t-test, Kruskal-Wallis test, Mann-Whitney U tests and Pearson's correlation coefficient, were conducted using IBM SPSS Statistics version 20 (IBM, Armonk, New York). Statistically significant features were further examined with post-hoc tests (Tukey-Kramer) to determine which groups of profiles differed from each other. Non-normal distribution data of two genera (*Anaerostipes* and *Escherichia-*

*Shigella*) in association with CKD were log-transformed into normal distributions. Statistical significance and 95% confidence intervals (95%CI) were calculated and considered as $P < 0.05$.

## Results

### Study population characteristics

Demographic, socioeconomic and clinical characteristics of CKD patients with and without *S. stercoralis* infection were matched (Table 1). No significant differences in these characteristics were found between the Ss+ and Ss- groups.

### Characterization of bacterial diversity and community structure

In total, 1551 OTUs were identified based on the 97%-similarity rule, with an average of 477 OTUs per sample. Sequences were classified into 16 bacterial phyla, 26 classes, 45 orders, 72 families and 258 genera and 189 species (including unidentified species). The species accumulation curves showed a saturation phase (Fig 1). This indicates that the sample size was sufficient to capture the overall microbiota structure.

The unweighted UniFrac distances, reflecting beta diversity, were significantly greater in the Ss- group than the Ss+ group ($P = 0.00019$). In terms of alpha diversity overall, there were no significant differences in estimated OTU richness, Chao1, the ACE metric, the Shannon diversity index and Good's coverage, ($P > 0.05$) (Table 2). In contrast, the alpha diversity in

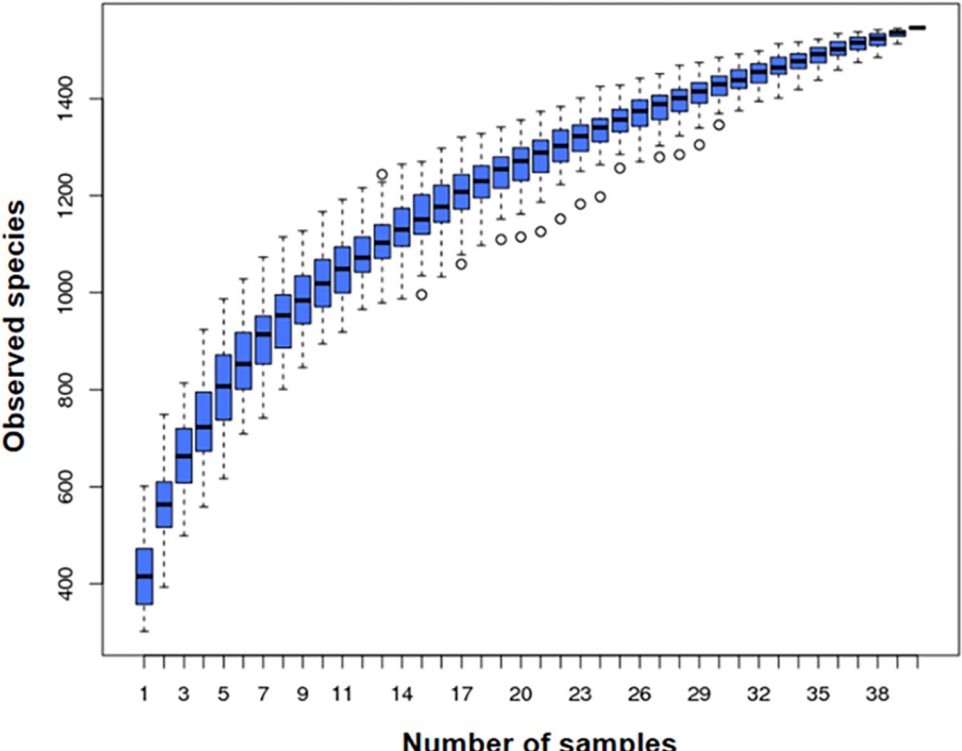

**Fig 1. Species accumulation curve.** X-axis: Number of samples, Y-axis: number of OTUs. Following an initial sharp rise in the number of OTUs as number of samples increases, there is a levelling of the plot. The narrow spread of the boxplots as the total number of samples is approached indicates that the number of samples was adequate to capture most of the microbial diversity present.

**Table 2. Alpha diversity of the gut microbiota in Ss- and Ss+ groups, calculated according to several indices.**

| Group | No. of Reads | No. of OTUs | Good's (%) | ACE | Chao 1 | PD whole tree | Shannon | Simpson |
|---|---|---|---|---|---|---|---|---|
| Ss- | 73173.3 | 483.7 | 0.99765 | 515.713 | 517.886 | 32.7308 | 5.25775 | 0.9182 |
| Ss+ | 74770.1 | 470.15 | 0.9976 | 507.982 | 504.266 | 32.5759 | 4.82045 | 0.88145 |
| P | | 0.45[a] | 0.706[b] | 0.78[a] | 0.76[b] | 0.92[a] | 0.08[b] | 0.218[b] |

Independent t-tests (a) and Mann-Whitney U tests (b) were used to calculate *P* values.

Abbreviations: OTU: Operational taxonomic units; ACE: Abundance-based coverage estimator; PD: Phylogenetic diversity

males in the Ss- group was significantly higher than in males in the Ss+ group (Shannon diversity index, *P* = 0.015; Simpson diversity index, *P* = 0.058) (Fig 2B and 2C).

The principal coordinate analysis (PCoA) was used to illustrate the beta diversity based on the unweighted UniFrac distances (Fig 2D). PCoA analysis revealed that the gut microbiota of Ss+ subjects deviated from the Ss- group (Fig 2E).

The LDA score showed a significant difference in abundance of certain taxa between the two groups. The candidate biomarker for the Ss- category was order Bradymonadales and for CKD with *S. stercoralis* infection (Ss+) was species *E. coli*, genus *Escherichia-Shigella* as well as the genus *Dialister*, family Veillonellaceae and order Selenomonadales (Fig 3).

## Differences in bacterial abundance between the Ss+ and Ss- groups

Proportions of sequence reads were compared between groups at the phylum and genus levels using Metastats. At the phylum level, there were no significant differences (Fig 4). For example, relative abundances of some principal taxa were: Firmicutes (Ss- 64.14% vs. Ss+ 59.33%; *P* = 0.39), Proteobacteria (Ss- 15.01% vs. Ss+ 18.83%; *P* = 0.50), Bacteroidetes (Ss- 12.61% vs. Ss+ 12.68%; *P* = 0.98), Actinobacteria (Ss- 3.46% vs. Ss+ 4.34%; *P* = 0.67) and Fusobacteria (Ss-

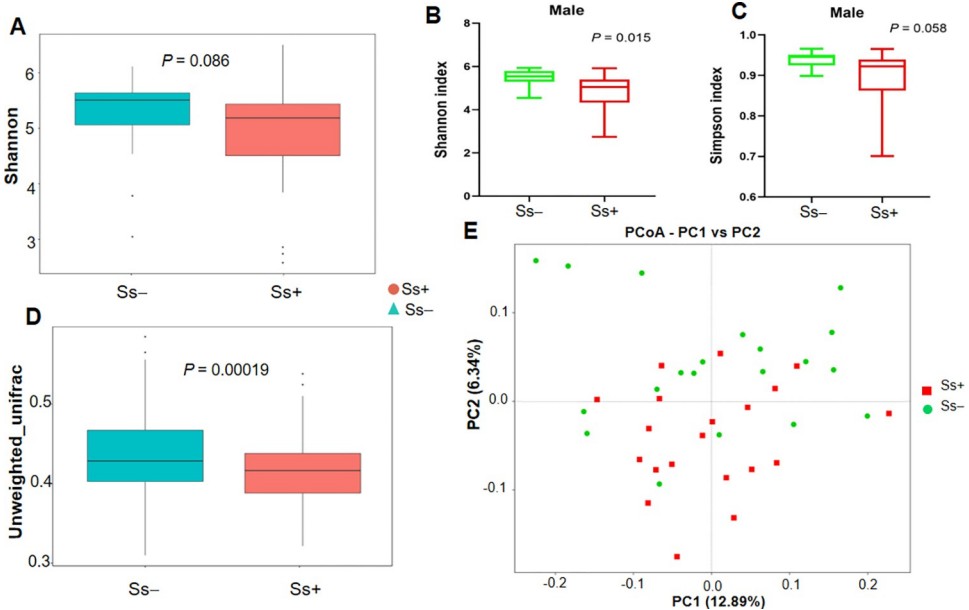

**Fig 2. Comparison of alpha diversity indexes and beta diversity in CKD patients with and without *S. stercoralis* infection.** (A) Shannon index (B) Shannon index in males (C) Simpson index in males. (D) Boxplot based on unweighted UniFrac distance. (E) Principal coordinate analysis (PCoA).

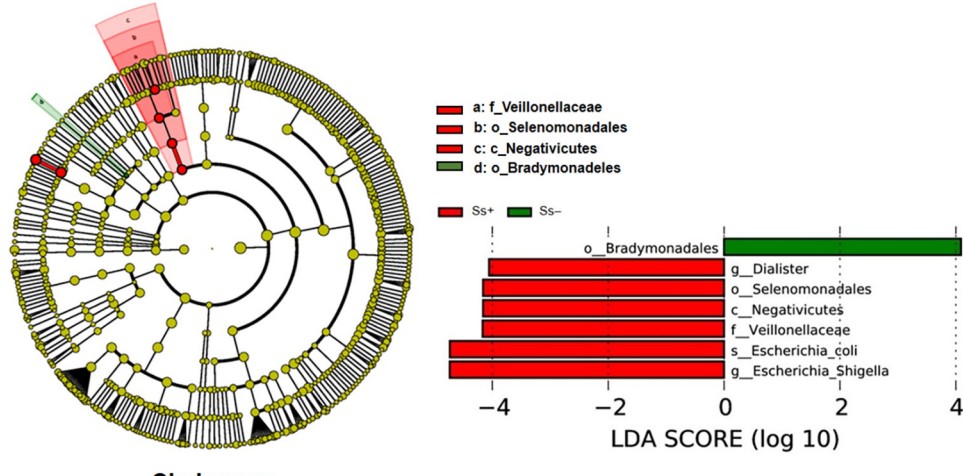

**Fig 3. Histogram of cladogram and linear discriminant analysis (LDA) score.** The histogram of the LDA scores presents taxa (potential biomarkers) whose abundance differed significantly among groups (Ss+ vs. Ss-) order Bradymonadales in Ss- (green color). Species *E. coli* belongs to the genus *Escherichia-Shigella*; genus *Dialister* belongs to the order Selenomonadales, class Negativicutes and family Veillonellaceae in Ss+ (red color). The cladogram shows specific taxa relevant to Ss+ and Ss- in the red and green nodes. The highest taxonomic level is towards the center of the diagram. The diameter of each circle represents the relative abundance of the taxon.

1.88% vs. Ss+ 3.81%; *P* = 0.52). However, at the genus level, 42 taxa were differentially relative abundance (Table 3).

To study the similarity among different samples, clustering analysis was applied. The unweighted pair group method with arithmetic mean (UPGMA), a type of hierarchical clustering method widely used in ecology. This showed that Ss- vs. Ss+ samples tended to cluster separately (Fig 5).

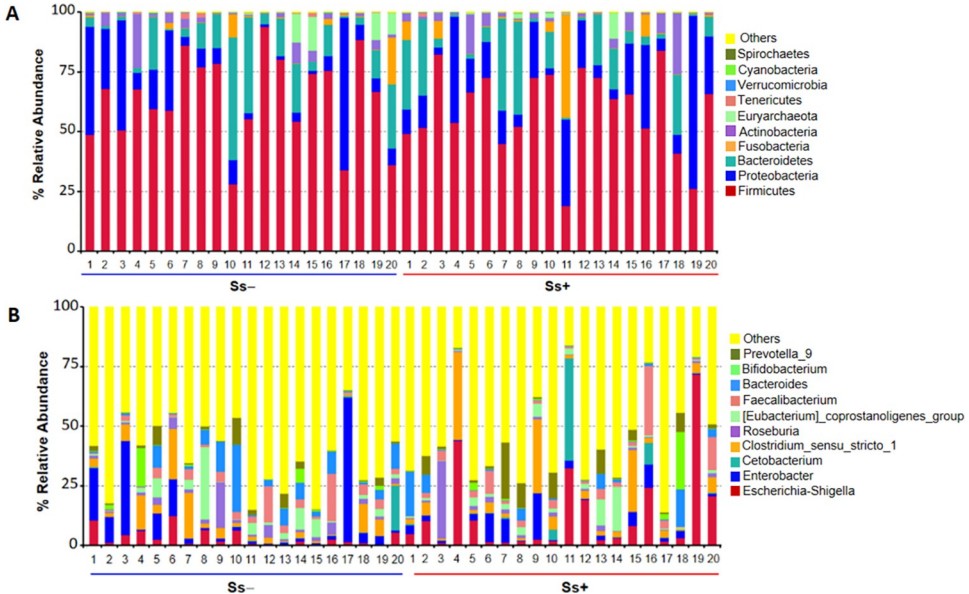

**Fig 4. The gut microbiota composition.** (A) and (B), Control group compared with *S. stercoralis*-infection group (Ss +) at the phylum and genus levels, respectively.

**Table 3. Taxa in the gut microbiome differing significantly between CKD patients with and without *S. stercoralis*.**

| Phylum | Family | Genus | Change in abundance | |
|---|---|---|---|---|
| | | | Ss+ | Ss- |
| Actinobacteria | Coriobacteriaceae | *Atopobium* | ↑ | |
| | | *Coriobacteriaceae_UCG-003* | ↑ | |
| | | *Gordonibacter* | | ↑ |
| | | *unidentified_Coriobacteriaceae* | | ↑ |
| | Corynebacteriaceae | *Corynebacterium* | ↑ | |
| | Micrococcaceae | *Rothia* | ↑ | |
| Bacteroidetes | Porphyromonadaceae | *Petrimonas* | ↑ | |
| | | *Proteiniphilum* | | ↑ |
| | Prevotellaceae | *Paraprevotella* | | ↑ |
| | | *Prevotella_1* | ↑ | |
| Cyanobacteria | unidentified_Gastranaerophilales | *unidentified_Gastranaerophilales* | | ↑ |
| Firmicutes | Carnobacteriaceae | *Lacticigenium* | ↑ | |
| | Leuconostocaceae | *Leuconostoc* | | ↑ |
| | Christensenellaceae | *Christensenella* | | ↑ |
| | Eubacteriaceae | *Anaerofustis* | | ↑ |
| | Family_XI | *Gallicola* | ↑ | |
| | | *Peptoniphilus* | ↑ | |
| | | *Tissierella* | ↑ | |
| | Lachnospiraceae | *[Eubacterium]_xylanophilum_group* | | ↑ |
| | | *Anaerosporobacter* | | ↑ |
| | | *Anaerostipes* | | ↑ |
| | | *Coprococcus_1* | | ↑ |
| | | *Lachnospiraceae_UCG-010* | ↑ | |
| | | *Tyzzerella_3* | | ↑ |
| | Peptostreptococcaceae | *Paeniclostridium* | | ↑ |
| | Ruminococcaceae | *Pseudoflavonifractor* | | ↑ |
| | | *Ruminiclostridium_1* | ↑ | |
| | | *Ruminococcaceae_UCG-011* | ↑ | |
| | Erysipelotrichaceae | *Erysipelotrichaceae_UCG-003* | | ↑ |
| | | *Erysipelotrichaceae_UCG-004* | | ↑ |
| | | *unidentified_Erysipelotrichaceae* | | ↑ |
| | Veillonellaceae | *Dialister* | ↑ | |
| Fusobacteria | Leptotrichiaceae | *Leptotrichia* | ↑ | |
| Proteobacteria | Neisseriaceae | *Eikenella* | ↑ | |
| | Rhodocyclaceae | *Dechlorobacter* | ↑ | |
| | Desulfobulbaceae | *Desulfobulbus* | ↑ | |
| | Cardiobacteriaceae | *Cardiobacterium* | ↑ | |
| | Enterobacteriaceae | *Cronobacter* | | ↑ |
| | Enterobacteriaceae | *Escherichia-Shigella* | ↑ | |
| | Pasteurellaceae | *Actinobacillus* | ↑ | |
| | Pasteurellaceae | *Aggregatibacter* | ↑ | |
| | Xanthomonadaceae | *Arenimonas* | ↑ | |

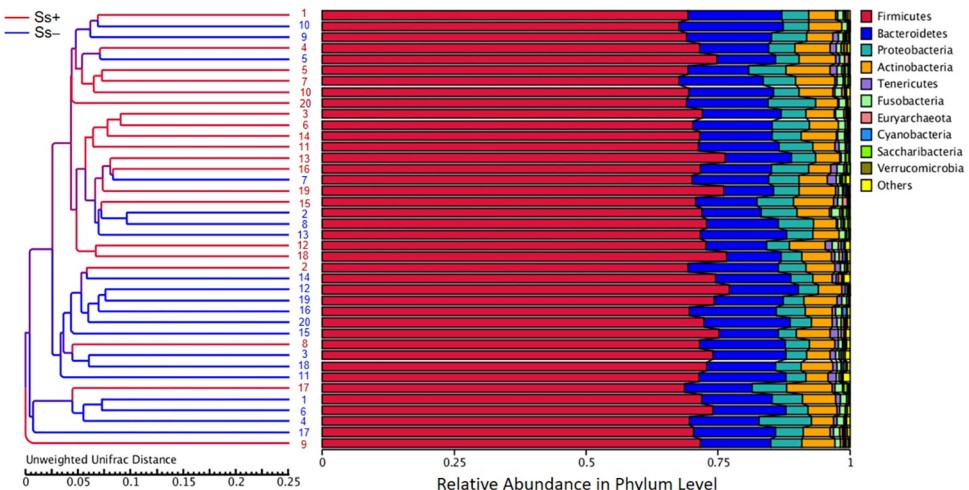

**Fig 5. Clustering using the unweighted pair group method with arithmetic mean (UPGMA).** UPGMA cluster tree based on unweighted UniFrac distances between CKD patients with or without *S. stercoralis* infection. The red branches represent individuals with *S. stercoralis* infection (Ss+) and the dark blue branches indicate uninfected (Ss-) individuals.

## The trends of some bacteria in *S. stercoralis* infection

Fig 6 shows the comparisons of abundance of some bacteria between Ss- and Ss+ groups at the genus level. Pathogenic taxa were more abundant in the Ss+ group and included genera such as: *Escherichia-Shigella* (3.36% vs. 13.33%; $P < 0.05$), *Streptococcus* (0.97% vs. 2.18%; $P > 0.05$), *Haemophilus* (0.46% vs 0.71%; $P > 0.05$), *Rothia* (0.024% vs. 0.11%; $P < 0.05$), *Actinomyces* (0.038% vs. 0.067%; $P > 0.05$) and *Aggregatibacter* (0.0013% vs. 0.025%; $P < 0.05$). Reduction of some SCFA-producing bacteria in the Ss+ group was observed, including *Eubacterium rectale*_group (4.51% vs. 3.78%; $P > 0.05$), *Eubacterium hallii*_group (1.24% vs. 0.94%; $P > 0.05$), *Anaerostipes* (0.54% vs. 0.074%; $P < 0.05$), *Coprococcus*_1 (0.11% vs. 0.057%; $P < 0.05$) and *Akkermansia* (0.081% vs. 0.043%; $P > 0.05$).

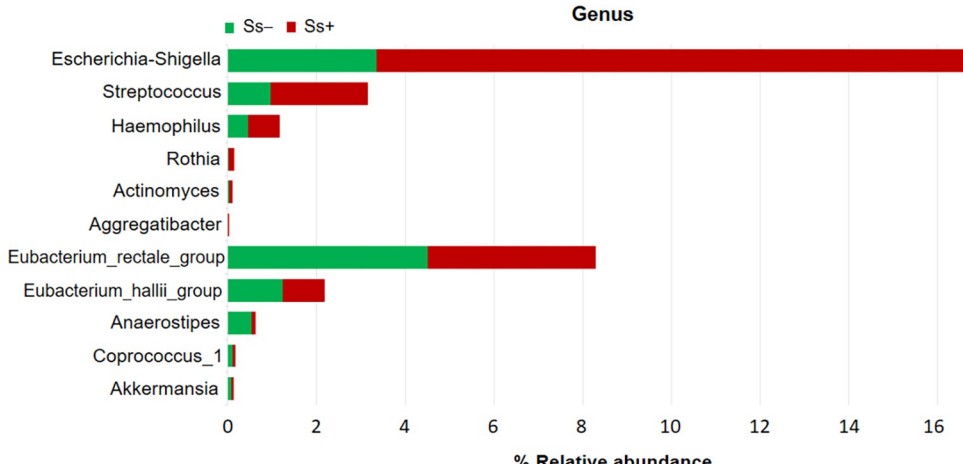

**Fig 6. Comparisons of abundance (numbers of sequence reads) of some bacteria between Ss- and Ss+ group.** Pathogenic bacteria: *Escherichia-Shigella*, *Streptococcus*, *Haemophilus*, *Rothia*, *Actinomyces*, *Aggregatibacter*. SCFA-producing bacteria: *Eubacterium rectale_group*, *Eubacterium hallii_group*, *Anaerostipes*, *Coprococcus_1*, *Akkermansia*.

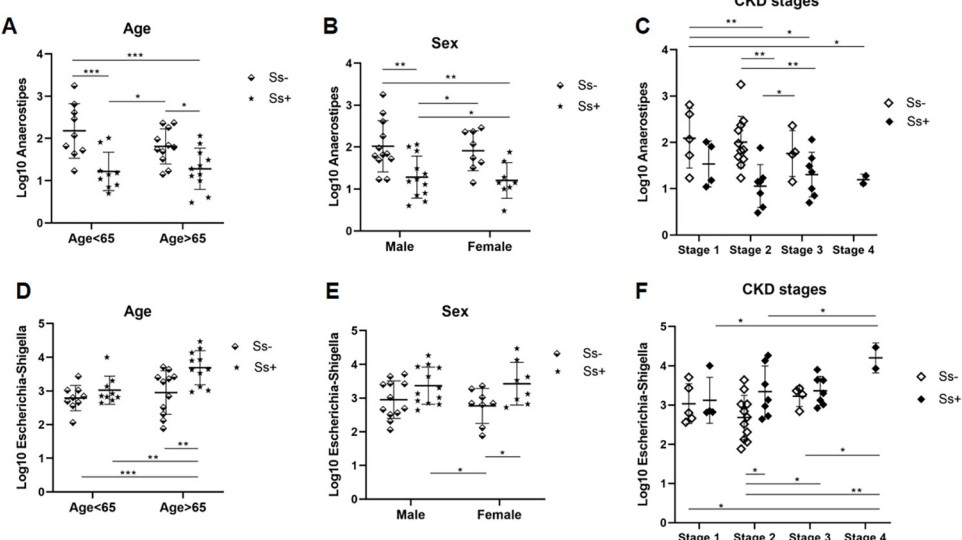

**Fig 7. Opposing trends in abundance of two genera, *Anaerostipes* and *Escherichia-Shigella*.** (A) and (D) Different trends related to age; (B) and (E) sex; (C) and (F) CKD stages. $^*P < 0.05$, $^{**}P < 0.01$, $^{***}P < 0.001$. Analysis of the difference among groups of sex, age and CKD stages based on one-way ANOVA test.

### *Anaerostipes* and *Escherichia-Shigella* exhibit opposing trends in abundance and correlate with sex, age and CKD stage

Fig 7 shows the opposing trends in abundance in the genera *Anaerostipes* and *Escherichia-Shigella*. The proportion of reads of *Anaerostipes* was lower in those aged over 65 years (Fig 7A), in females (7B), and with increasingly advanced CKD stage (7C) and in those infected with *S. stercoralis* (Fig 7A, 7B and 7C). The opposite was observed in the case of *Escherichia-Shigella*: higher proportions of sequence reads of this genus were seen in elderly (>65 years) individuals (Fig 7D), in females (Fig 7E) and in those infected with *S. stercoralis* (Fig 7D, 7E and 7F). More reads of *Escherichia-Shigella* were seen with increasingly advanced CKD stage (Fig 7F).

## Discussion

In this study, we first characterized the gut microbiota of CKD patients with and without *S. stercoralis* using high-throughput sequencing of the V3–V4 region of the 16S rRNA gene. The results showed that *S. stercoralis* infection altered gut microbiota composition in CKD patients, leading to lower microbial diversity. This study also suggested that microbial candidate biomarkers for CKD concurrent with *S. stercoralis* infection include *Escherichia coli* (genus *Escherichia-Shigella*, phylum Proteobacteria) and the genus *Dialister* (family Veislonellaceae, order Selenomonadales, class Negativicutes, phylum Firmicutes).

Various parameters including gender, age and other factors have been reported to affect the gut microbiota [37]. With this in mind, we matched Ss+ and Ss- subjects for these parameters to reduce confounding factors. This allowed us to identify changes in the gut microbiome due to infection with *S. stercoralis* in CKD patients. Our results revealed that the alpha-diversity indices (Chao1, the ACE metric, the Shannon diversity index, Good's coverage) did not significantly differ between the two groups. However, the Shannon diversity index in males (n = 12) in Ss+ group was significantly lower than in the Ss- group. In addition, the beta diversity, based on the unweighted UniFrac distances, was significantly lower in the Ss+ group, suggesting a decrease in ecological diversity in CKD concurrent with *S. stercoralis* infection. These

slight differences spanned all taxonomic levels of the microbiota. At the phylum level, abundance of Firmicutes was reduced while abundance of Proteobacteria and Fusobacteria increased in Ss+ subjects. At the family level, there was an increase of Clostridiaceae, Streptococcaceae, Desulfovibrionaceae and Enterobacteriaceae in the Ss+ group. A previous study demonstrated that these families were associated with trimethylamine (TMA) production [38]. The high abundances of families Enterobacteriaceae, Clostridiaceae and Veillonellaceae in Ss + subjects are in agreement with a previous study [39]. These bacteria are associated with increasing fecal pH [39] to a level where most opportunistic bacterial pathogens prefer to grow [40]. The change of microbiota composition that we observed may influence the environment in the gut, suggesting that *S. stercoralis* infection may influence the microbiota and modulate the pH of the gut environment.

Forty-two genera showed contrasting abundances between the two groups. Interestingly, there were significant increases of pathogenic bacteria including *Escherichia-Shigella*, *Rothia* and *Aggregatibacter* and some increase of *Actinomyces*, *Streptococcus* and *Haemophilus* in CKD patients infected with *S. stercoralis* compared to uninfected controls. In contrast, significant reduction of some SCFA-producing bacteria, such as *Anaerostipes* and *Coprococcus*_1 and some decrease of *Akkermansia*, *Eubacterium rectale*_group and *Eubacterium hallii*_group were noted in the Ss+ group. Specifically, abundance of the genus *Escherichia-Shigella* is known to be positively correlated with uremic toxins such as trimethylamine-N-oxide and indoxyl sulfate [41,42]. Our results demonstrated a significant inverse correlation of *Escherichia-Shigella* with the estimated glomerular filtration rate (eGFR r = -0.37, *P* = 0.018). eGFR is one criterion for diagnosis and staging of CKD. Thus, high abundance of *Escherichia-Shigella* was correlated with low eGFR value and higher CKD stage. In addition, *Enterobacteriaceae* and *E. coli* are markedly more abundant in individuals with impaired kidney function as demonstrated previously [43], highlighting that there is an association between the genus *Escherichia-Shigella* and CKD with concurrent *S. stercoralis* infection.

Interestingly, the genus *Anaerostipes* was less abundant in CKD patients with concurrent *S. stercoralis* infection than in those without. Members of this genus are Gram-variable, obligate anaerobes which produce acetate, butyrate and lactate from glucose fermentation [44]. Our findings were partially consistent with those of a previous study, which found that *Anaerostipes* had low relative abundance in CKD in an animal model and noted that this genus was negatively correlated with amount of intestinal urea. Nephrectomized mice with low levels of *Anaerostipes* exhibited negative effects on kidney parameters (BUN and creatinine) [45]. However, we found no correlation here between *Anaerostipes* and kidney parameters. This may be due to the limited sample size. Recent research indicated that elevated levels of *Anaerostipes* led to increased eGFR and improvement in renal function [46]. Furthermore, the relative abundance of the genus *Anaerostipes* was markedly reduced in nonsurvivors with end-stage kidney disease (ESKD) [47]. Specifically, we found that one species, *Anaerostipes hadrus*, an important microbe in maintaining intestinal metabolic balance [48], was significantly reduced in CKD with concurrent *S. stercoralis* infection.

A previous study using a rat model revealed that increased levels of *Rothia* were positively associated with creatinine levels in acute kidney injury and with severity of kidney damage [49]. *Rothia* spp. are Gram-positive cocco-bacilli that cause a wide range of serious infections, especially in immunocompromised hosts. *Rothia* is often identified in blood cultures from patients with bacteremia [50]. In this study, we found a positive correlation between *Rothia* and the genus *Streptococcus* (r = 0.47, P = 0.002). A similar relationship was observed in a recent study, which found that the log-ratio between the presence of the genera *Rothia* and *Streptococcus* was the best predictor of creatinine level [49].

The main SCFA-producing bacteria in humans [51] including *Faecalibacterium prausnitzii* (butyrate-producing bacteria in the phylum Firmicutes), *Eubacterium rectale* and *E. hallii*

(family Lachnospiraceae) and *Anaerostipes* spp. (sugar/lactate-utilizing bacteria producing butyrate from lactate and acetate) were all less abundant in *S. stercoralis* infection. A previous study observed a decrease in the level of serum SCFAs in CKD patients and an inverse correlation between butyrate level and renal function [52]. Our study suggested that SCFA-producing bacteria are depleted in CKD with concurrent *S. stercoralis* infection, which may affect CKD progression. However, further studies are needed to confirm this association.

Our study has strengths and limitations. An important strength of this study is that we used groups that were pair-matched for sex, age and biochemical factors. However, we did not obtain data for concentrations of uremic toxins (TMAO and IS) or for amounts of SCFAs. Moreover, we did not record the clinical manifestation of *S. stercoralis* infection in CKD patients so we were not able to show the association between some pathogenic bacteria and *S. stercoralis* infection in CKD. In addition, the sample size in this study was small due to the limited number of individuals in the population with CKD and infection with *S. stercoralis* only. The small sample sizes affect the estimation of microbiome alpha diversity and statistical power in analyses.

## Conclusions

This study suggests that *S. stercoralis* infection reduces the diversity of the gut microbiota in CKD patients. An increased abundance of harmful bacteria and reduction of some SCFA-producing bacteria in *S. stercoralis* infection was found. In addition, the abundance of members of the genus *Escherichia-Shigella* was significantly and inversely correlated with eGFR levels. Significant elevation of members of this genus in CKD patients with *S. stercoralis* infection may indicate potential diagnostic markers for CKD in *S. stercoralis*-endemic areas. Thus, we suggest that these changes in the composition of the gut microbiome in *S. stercoralis* infection may result in disruption of the gut barrier structure and absorption of harmful products that can contribute to toxicity, inflammation and malnutrition, contributing to CKD progression. Future metabolomics studies are required to unravel the relationship between CKD and *S. stercoralis* infection.

## Acknowledgments

N.T.H. thanks the scholarship under the Doctoral Training Program from Research Affairs, Medicine Faculty and Graduate School, Khon Kaen University, Thailand. We would like to thank to all of the people who voluntarily participated in this study. We also acknowledge Prof. David Blair from Publication Clinic KKU, Thailand, for his comments and editing the manuscript.

## Author Contributions

**Conceptualization:** Nuttanan Hongsrichan, Porntip Pinlaor, Somchai Pinlaor.

**Formal analysis:** Nguyen Thi Hai, Nuttanan Hongsrichan, Kitti Intuyod, Porntip Pinlaor, Manachai Yingklang, Apisit Chaidee, Thatsanapong Pongking, Sirirat Anutrakulchai, Ubon Cha'on, Somchai Pinlaor.

**Funding acquisition:** Nguyen Thi Hai, Sirirat Anutrakulchai, Somchai Pinlaor.

**Investigation:** Nguyen Thi Hai, Nuttanan Hongsrichan, Kitti Intuyod, Porntip Pinlaor, Manachai Yingklang, Apisit Chaidee, Thatsanapong Pongking, Sirirat Anutrakulchai, Ubon Cha'on, Somchai Pinlaor.

**Project administration:** Somchai Pinlaor.

**Resources:** Sirirat Anutrakulchai, Ubon Cha'on, Somchai Pinlaor.

**Supervision:** Nuttanan Hongsrichan, Porntip Pinlaor, Somchai Pinlaor.

**Visualization:** Nguyen Thi Hai, Kitti Intuyod, Apisit Chaidee, Thatsanapong Pongking.

**Writing – original draft:** Nguyen Thi Hai.

**Writing – review & editing:** Nguyen Thi Hai, Nuttanan Hongsrichan, Somchai Pinlaor.

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
