## [Decision Letter · Decision Letter 0]

28 Jun 2022

Dear Dr Pinlaor,

Thank you very much for submitting your manuscript "Strongyloides stercoralis infection induces gut dysbiosis in chronic kidney disease patients" for consideration at PLOS Neglected Tropical Diseases. As with all papers reviewed by the journal, your manuscript was reviewed by members of the editorial board and by several independent reviewers. The reviewers appreciated the attention to an important topic. Based on the reviews, we are likely to accept this manuscript for publication, providing that you modify the manuscript according to the review recommendations. 

Sincerely,

Alessandra Morassutti, PhD

Associate Editor

Abhay Satoskar

Deputy Editor

Reviewer's Responses to Questions

**Key Review Criteria Required for Acceptance?**

**Methods**

-Are the objectives of the study clearly articulated with a clear testable hypothesis stated?

-Is the study design appropriate to address the stated objectives?

-Is the population clearly described and appropriate for the hypothesis being tested?

-Is the sample size sufficient to ensure adequate power to address the hypothesis being tested?

-Were correct statistical analysis used to support conclusions?

-Are there concerns about ethical or regulatory requirements being met?

Reviewer #1: A clear testable hypothesis "that S. stercoralis infection changes gut microbiome, contributing to progression of chronic kidney disease" is stated. The study design and population are clearly described. This study compared the bacterial composition of the faecal microbiome of infection-driven chronic kidney disease CKD patients who were infected with Strongyloides stercoralis Ss+ with that of those who were uninfected with S. stercoralis Ss-. The groups were pair-matched for sex, age and biochemical factors. 

PCR testing was undertaken to ensure that the “uninfected” group was free from S. stercoralis infection. 

The microbial composition of the faeces of each patient was characterised by the V3-V4 region of the 16SrRNA. A sequencing library was generated for each sample. These were subjected to data control processes ensuring the quality of the data. 

The sample size was limited by the availablity of suitable participants. Nevertheless, given that the Ss+ patients were matched with Ss- patients, meaningful results were obtained.

Suitable statistical analysis was used to support the conclusions.

The manuscript did not indicate whether informed consent had been obtained from the participants.

**Results**

-Does the analysis presented match the analysis plan?

-Are the results clearly and completely presented?

-Are the figures (Tables, Images) of sufficient quality for clarity?

Reviewer #1: The analysis presented matched the analysis plan. The results were clearly presented and the figures of sufficient quality for clarity.

The data showed a relationship between S. stercoralis infection and altered microbial composition in CKD patients.

258 genera from 16 phyla were present in the total group. Overall, alpha diversity was similar in the two groups, but in males, alpha diversity was significantly greater in the Ss- group. Beta diversity was also significantly greater in the Ss- group.

At the phylum level, there was no significant difference in abundance between the Ss+ and Ss- groups, but at the genus level 42 taxa differed in relative abundance. Pathogenic genera Escherichia-Shigella, Steptococcus, Haemophilus, Rothia, Actinomycetes, Aggregatibacter were significantly increased in the Ss+ group. Short chain fatty acids- SCFA-producing bacteria Eubacterium rectale_group, Eubacterium hallii_group, Anaerostipes, Coprococcus and Akkermansia were significantly decreased in the Ss+ group.

The abundance Anaerostipes was significantly lower in those aged over 65 years, in females, in increasingly advanced CKD stage, and in those infected with S. stercoralis. The abundance of Escherichia-Shigella was significantly higher in those aged over 65 years, in females, in increasingly advanced CKD stage and in those infected with S. stercoralis.

**Conclusions**

-Are the conclusions supported by the data presented?

-Are the limitations of analysis clearly described?

-Do the authors discuss how these data can be helpful to advance our understanding of the topic under study?

-Is public health relevance addressed?

Reviewer #1: The conclusions are supported by the data presented and the limitations are clearly described.

The authors conclude that the changes in the composition of the microbiome in S. stercoralis infections may result in disruption of the gut barrier that can contribute to toxicity, inflammation and malnutrition and progression of CKD, and its public health importance in areas endemic for S. stercoralis. They also indicate what kind of studies are needed to further elucidate this relationship between S. stercoralis infection and CKD.

**Editorial and Data Presentation Modifications?**

Reviewer #1: Minor issues

Line 31: In the abstract, “infection-driven” chronic kidney disease is mentioned, but this is the only time “infection-driven” is mentioned in the manuscript. Would you please clarify whether this paper is about infection-driven CKD or CKD of any aetiology. If genuinely “infection-driven” CKD, please clarify in the author summary and the manuscript eg in line 116.

Line 125: renal infection with other intestinal parasites. Should this be “renal infection, infection with other parasites.”?

In Table 1, eosinophilia is elevated in both the Ss+ and Ss- groups, which suggests that there were other helminth infections in the Ss- group. Would you please explain this anomaly?

Line 208: Is the number of species correct here? The number should be more than or equal to the number of genera.

Line 211-212: State in the text which group had the greater beta diversity [the Ss- group, according to Fig 2].

Minor grammatical, spelling or punctuation changes:

Line 72: change “significant” to “significantly”

Line 116: change “age) of” to “age) with”

Line 220: change “significantly” to “significance”

Line 306: change “Anearostipes” to “Anaerostipes”

Line 583 (Caption for Fig 3): change “Oder” to “Order”

Line 618 (Caption for Fig 7): insert “;” after “age” 

Recommendation: Minor Revision

**Summary and General Comments**

Reviewer #1: To my knowledge, this study is the first to show in CKD patients that S. stercoralis infection is associated with decreased diversity of the faecal microbiota, increased abundance of pathogenic microbial genera and decreased abundance of SCFA-producing genera. These are very important findings.

Although the study involved a small number of participants, the use of matched pairs ensured that the S. stercoralis-infected group was similar to the uninfected group in characteristics other than S. stercoralis infection status. PCR testing was used to ensure as much as possible that the uninfected group was truly S. stercoralis-free. The authors were careful to ensure that there was sufficient DNA in the extracted samples to carry out gene sequencing.

Although the manuscript states that the study protocol was approved by the human ethical review committee of Khon Kaen University (HE631200), the manuscript does not state whether informed consent was obtained from the participants. Details about informed consent should be included in the manuscript.

PLOS authors have the option to publish the peer review history of their article (what does this mean?). If published, this will include your full peer review and any attached files.

Reviewer #1: Yes: Jennifer Shield

Figure Files:

Data Requirements:

Reproducibility:

References

---

## [Decision Letter · Decision Letter 1]

9 Aug 2022

Dear Somchai Pinlaor, We are pleased to inform you that your manuscript 'Strongyloides stercoralis infection induces gut dysbiosis in chronic kidney disease patients' has been provisionally accepted for publication in PLOS Neglected Tropical Diseases.

Best regards,

Alessandra Morassutti, PhD

Academic Editor

Abhay Satoskar

Section Editor

Reviewer's Responses to Questions

**Key Review Criteria Required for Acceptance?**

**Methods**

-Are the objectives of the study clearly articulated with a clear testable hypothesis stated?

-Is the study design appropriate to address the stated objectives?

-Is the population clearly described and appropriate for the hypothesis being tested?

-Is the sample size sufficient to ensure adequate power to address the hypothesis being tested?

-Were correct statistical analysis used to support conclusions?

-Are there concerns about ethical or regulatory requirements being met?

Reviewer #1: This is a re-review of the manuscript with minor modifications.

The study meets the requirements listed.

**Results**

-Does the analysis presented match the analysis plan?

-Are the results clearly and completely presented?

-Are the figures (Tables, Images) of sufficient quality for clarity?

Reviewer #1: The analysis and the presentation of results are clearly and completely presented, the figures are appropriate and clear.

**Conclusions**

-Are the conclusions supported by the data presented?

-Are the limitations of analysis clearly described?

-Do the authors discuss how these data can be helpful to advance our understanding of the topic under study?

-Is public health relevance addressed?

Reviewer #1: The conclusions are supported by the data presented, the limitations are clearly described, the authors discuss the relevance of the results to our understanding of CKD, Strongyloides infection and impact on the microbiota. The public health relevance is addressed.

**Editorial and Data Presentation Modifications?**

Reviewer #1: Accept

**Summary and General Comments**

Reviewer #1: To my knowledge, this study is the first to show in CKD patients that S. stercoralis infection is associated with decreased diversity of the faecal microbiota, increased abundance of pathogenic microbial genera and decreased abundance of SCFA-producing genera. These are very important findings.

Although the study involved a small number of participants, the use of matched pairs ensured that the S. stercoralis-infected group was similar to the uninfected group in characteristics other than S. stercoralis infection status. PCR testing was used to ensure as much as possible that the uninfected group was truly S. stercoralis-free. The authors were careful to ensure that there was sufficient DNA in the extracted samples to carry out gene sequencing.

This is an important study, and I look forward to its publication.

PLOS authors have the option to publish the peer review history of their article (what does this mean?). If published, this will include your full peer review and any attached files.

Reviewer #1: **Yes: **Jennifer Shield

---

## [Editor Report · Acceptance letter]

31 Aug 2022

Dear Professor Pinlaor,

We are delighted to inform you that your manuscript, "*Strongyloides stercoralis* infection induces gut dysbiosis in chronic kidney disease patients," has been formally accepted for publication in PLOS Neglected Tropical Diseases.

Best regards,

Shaden Kamhawi

co-Editor-in-Chief

Paul Brindley

co-Editor-in-Chief
